# A comprehensive characterization of PncA polymorphisms that confer resistance to pyrazinamide

Adam N. Yadon[1,2], Kashmeel Maharaj[2], John H. Adamson[2], Yi-Pin Lai[3], James C. Sacchettini[4], Thomas R. Ioerger[3], Eric J. Rubin[1] & Alexander S. Pym[2]

Tuberculosis chemotherapy is dependent on the use of the antibiotic pyrazinamide, which is being threatened by emerging drug resistance. Resistance is mediated through mutations in the bacterial gene *pncA*. Methods for testing pyrazinamide susceptibility are difficult and rarely performed, and this means that the full spectrum of *pncA* alleles that confer clinical resistance to pyrazinamide is unknown. Here, we performed in vitro saturating mutagenesis of *pncA* to generate a comprehensive library of PncA polymorphisms resultant from a single-nucleotide polymorphism. We then screened it for pyrazinamide resistance both in vitro and in an infected animal model. We identify over 300 resistance-conferring substitutions. Strikingly, these mutations map throughout the PncA structure and result in either loss of enzymatic activity and/or decrease in protein abundance. Our comprehensive mutational and screening approach should stand as a paradigm for determining resistance mutations and their mechanisms of action.

[1] Department of Immunology and Infectious Disease, Harvard T.H. Chan School of Public Health, 665 Huntington Ave., Bldg 1, Rm 810, Boston, MA 02115, USA. [2] African Health Research Institute (AHRI), Nelson R. Mandela School of Medicine, K-RITH Tower Building, Level 3, 719 Umbilo Road, Durban 4001, South Africa. [3] Department of Computer Science and Engineering, 3112 Texas A&M University, 301 H.R. Bright Building, College Station, TX 77843, USA. [4] Department of Biochemistry and Biophysics, Texas A&M University, Interdisciplinary Life Sciences Building, 301 Old Main Dr., College Station, TX 77843, USA. Adam N. Yadon and Kashmeel Maharaj contributed equally to this work. Correspondence and requests for materials should be addressed to E.J.R. (email: erubin@hsph.harvard.edu) or to A.S.P. (email: alex.pym@ahri.org)

Tuberculosis (TB) is a global public heath challenge that caused 1.4 million deaths in 2015[1]. Antibiotic chemotherapy is a mainstay of TB control programs, however, the emergence of drug resistance is a significant problem, with an estimated 580,000 cases in 2015[1]. Standard chemotherapy for TB requires a 6-month regimen of a combination of four antibiotics: rifampicin (RIF), isoniazid (INH), ethambutol, and pyrazinamide (PZA)[1, 2].

PZA is a cornerstone of current and future, TB treatment regimens[3–5]. Its introduction into TB chemotherapies was critical for reducing treatment to 6 months[6–9]. PZA is also under evaluation in new regimens designed to further reduce treatment duration and more effectively treat drug resistant TB[3–5, 10]. Despite its central role and ubiquitous use in TB therapy, PZA drug-susceptibility testing (DST) is not widespread[11], nor part of the World Health Organization's[12] recommendations. To improve treatment outcomes and reduce transmission, prompt diagnosis of PZA drug resistance is paramount. The development of rapid and reliable DST is therefore a priority for TB control.

The current gold-standard for PZA DST is a whole-cell phenotypic approach using the BD BACTEC MGIT 960 system[12]. Unfortunately, testing is notoriously challenging and unreliable, especially in clinical isolates[11, 13–20]. The limitations and inconsistencies result from the difficulty of growing TB under acidic (pH < 6.0) conditions required by the assay[8, 21], inoculum size effects (large inoculums increase false resistance rates)[8, 14, 22], and growth-phase-dependent activity (limited activity against rapidly growing bacilli)[6, 14].

Recently developed molecular genetic diagnostic technologies, such as GeneXpert (Cepheid, Sunnyvale, CA) and GenoType MTBDRplus (Hain Lifescience, Nehren, Germany) rapidly detect mutations that most frequently cause resistance to rifampin and INH[23–30]. Such assays rely on small genomic "hot-spot" regions where highly penetrant mutations directly correlate with phenotypic drug resistance. Such an assay for PZA would be extremely beneficial.

The primary drivers of PZA resistance are mutations in the *pncA* gene[11, 31–36]. PncA is a non-essential[37–41] intracellular pyrazinamidase (PZase) that converts PZA (a prodrug) to its active form, pyrazinoic acid (POA)[34, 42]. In clinical isolates, diverse *pncA* alleles, which include single-nucleotide and multi-nucleotide polymorphisms and indels, are found across the full 561 base-pair (bp) open-reading frame[11, 31–33]. As a result, no genetic "hot-spot" region comprising highly penetrant mutations has been identified. An attractive molecular diagnostic alternative is full gene sequencing. However, because PZA susceptibility testing is rare, the entire spectrum of *pncA* alleles conferring clinically significant PZA resistance has not been defined and polymorphisms in PncA have been found in susceptible isolates[11, 32, 33, 43]. Thus, a molecular diagnostic for PZA resistance is currently impractical and would require a comprehensive and systematic assessment of mutations conferring PZA resistance.

Here, we took an unbiased approach to assess the impact of all *pncA* single-nucleotide polymorphisms (SNPs) on PZA susceptibility. We generated a comprehensive library of *pncA* mutations in *Mycobacterium tuberculosis* (*Mtb*) using random PCR mutagenesis, and screened the pooled library for PZA resistance both in vitro and in infected mice. We identified a large repertoire of resistance-conferring substitutions, some previously seen in clinical isolates and, importantly, many that have not. We find

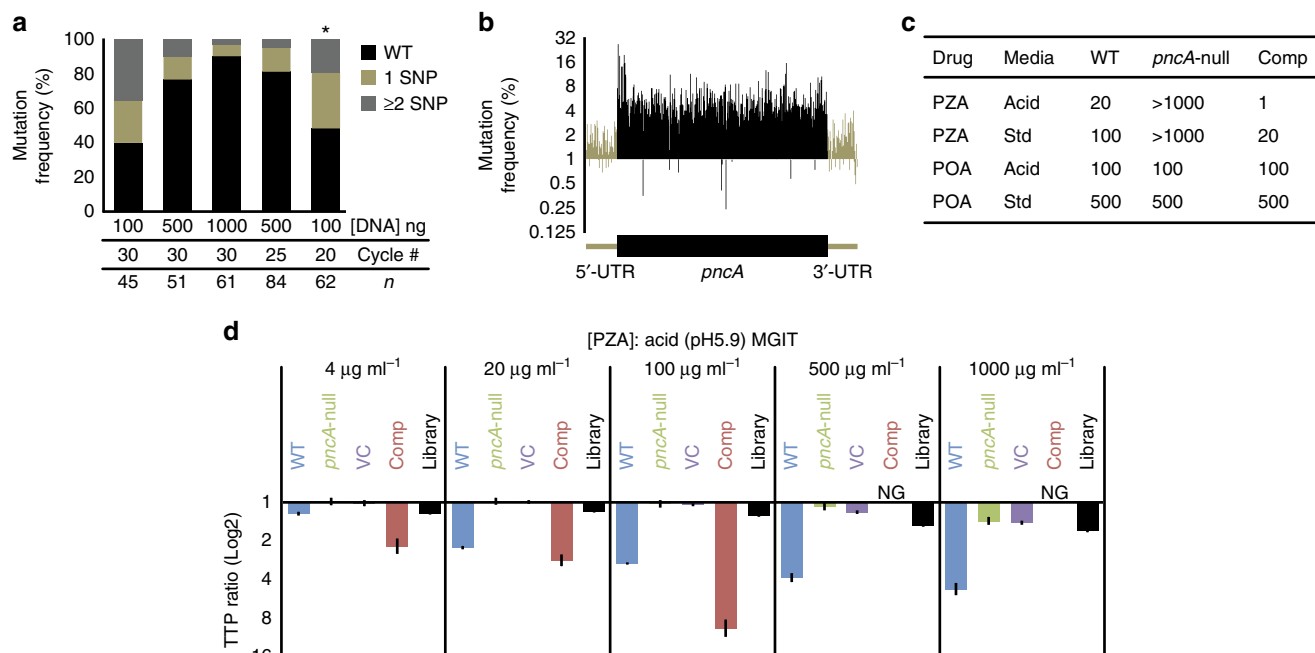

**Fig. 1** Generation of a comprehensive *pncA* mutant library in *M. tuberculosis*. **a** Frequency (%) of *pncA* single-nucleotide polymorphisms in five *pncA* libraries generated by random PCR mutagenesis using different template concentrations (DNA ng) and cycle numbers (#). *n* corresponds to the number of single colonies sequenced from each library and * the library selected for screening. **b** The mean *pncA* SNP frequency (%) from three biological replicates at each nucleotide of our candidate library relative to the mean of three biological replicates of an unmutagenized wild-type control determined using Illumina sequencing. **c** Minimum inhibitory concentration (*MIC*, μg ml⁻¹) of pyrazinamide (PZA) and pyrazinoic acid (POA) for three isogenic control strains, WT, *pncA*-null, and Comp, in pH 5.9 (Acid) and pH 6.8 (Std) media. **d** Growth inhibition of four isogenic control strains, WT, *pncA*-null, Comp and a vector control (*VC*), and the *pncA* library with increasing concentrations of pyrazinamide in pH 5.9 (acid) media. Time-to-positivity (*TTP*) ratio is the mean of the time-to-positivity of the test condition relative to the mean of the no-drug control. A minimum of three biological replicates per strain per test condition was performed. *Error bars* represent the standard deviations derived from the propagation of error using the quotient of the coefficient of variation from each condition. *NG* no growth

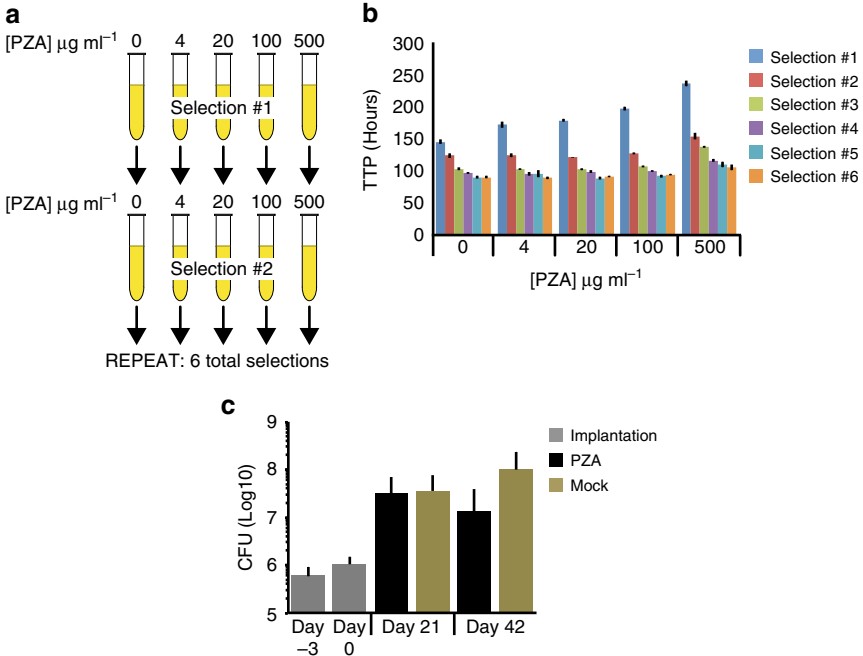

**Fig. 2** Pyrazinamide selection screens in vitro and during infection in mice. **a** Schematic diagram of the sequential in vitro pyrazinamide (*PZA*) selection screen. **b** Mean time-to-positivity (*TTP*; hours) of three biological replicates and standard deviations for each round of in vitro selection. **c** Mean and standard deviations of bacterial burdens (colony-forming units *CFU*) observed in the spleens of mice at day −3 (implantation; *gray*), day 0 (start of pyrazinamide treatment; *gray*) and after 21 and 42 days of pyrazinamide (*PZA*; *black*) or mock (*brown*) treatment. Five mice per time-point were analyzed

that resistance results largely from a loss of PncA enzymatic activity and/or protein abundance, which can be empirically determined but not easily predicted from the primary sequence. Furthermore, we find many substitutions that do not confer resistance to PZA. This catalog of PncA substitutions thus provides a comprehensive roadmap for the accurate molecular characterization of PZA resistance. Our comprehensive mutational and screening approach should stand as a paradigm for determining resistance mutations and their mechanisms of action.

## Results

**PncA mutant library**. To systematically assess the phenotype of all *pncA* SNPs, we constructed libraries of *pncA* mutants using error-prone, random PCR mutagenesis (Fig. 1a). A candidate library, containing the highest proportion of SNPs, was transformed into an H37Rv *pncA* null strain (*pncA*-null)[38]. The resultant library was characterized by Illumina sequencing of full-length *pncA* amplicons, with reads containing multiple mutations, or indels discarded. SNPs were markedly enriched within the coding region of *pncA* relative to an unmutagenized wild-type (WT) control (Fig. 1b), averaging 4.5-fold higher at each nucleotide. Therefore, despite the presence of WT *pncA* within our library, our sequencing was sufficiently sensitive to detect mutations above the background error rate throughout the gene.

**In vitro selection of pyrazinamide resistant substitutions**. As many factors influence the reproducibility of in vitro PZA DST we used the BD BACTEC MGIT 960 system[12] with standardized inoculums from titered frozen stocks (Supplementary Fig. 1). We first defined the minimum inhibitory concentrations (MICs) (Fig. 1c) for our control strains. In acid (pH 5.9) media, the *pncA*-null strain was highly resistant with a MIC of >1000 μg ml⁻¹. In contrast, a *pncA*-null strain complemented with a WT *pncA* gene (Comp) was hypersusceptible to PZA relative to the parental

H37Rv strain (WT), with MICs of 1 and 20 μg ml⁻¹, respectively. RT-qPCR confirmed a 4-fold over-expression of *pncA* (Supplementary Fig. 2)[44], consistent with our use of the strong mycobacterial optimized promoter (MOP)[45] to drive *pncA* transcription. The PZA hypersusceptibility in our complement strain increases the dynamic range in which PZA susceptibilities can be tested. To define this further we measured growth inhibition in the presence of increasing PZA concentrations (Fig. 1d). The growth of the Comp strain was inhibited even at the lowest concentration tested, with all growth suppressed at concentrations greater than 100 μg ml⁻¹. There were no growth differences in any strain after treatment with POA (Supplementary Fig. 3), indicating that PZA susceptibility is solely due to enzymatic activation by PncA. Growth inhibition was greatest in acid media (Fig. 1d and Supplementary Fig. 3) and was thus the conditions used for our in vitro screen.

To enrich for resistant clones, we exposed our pooled *pncA* mutant library to six sequential rounds of in vitro selection using a range of PZA concentrations (Fig. 2a). As expected, the rate of growth, determined by the time-to-positivity (TTP), increased with each successive selection (Fig. 2b and Supplementary Fig. 4), consistent with the enrichment of resistant clones. After Illumina sequencing of full-length *pncA* amplicons, the average mutation frequency at *pncA* nucleotides changed progressively after each successive PZA selection (Supplementary Fig. 5). Importantly, there was minimal variation in the average mutation frequency at *pncA* nucleotides in the absence of PZA (Supplementary Fig. 6), indicating that selection only occurred in the presence of drug.

We then identified polymorphisms under positive PZA selection across all drug concentrations and selections. We first excluded mutations that were both under-represented and showed no evidence of selection (Supplementary Fig. 7), yielding a library representation of ~88% (*N* = 977) of all non-synonymous amino acid substitutions resultant from a SNP (Supplementary Data 1 and 2). We thus identified a total of 264 enriched SNPs (Supplementary Data 1), resulting in 191

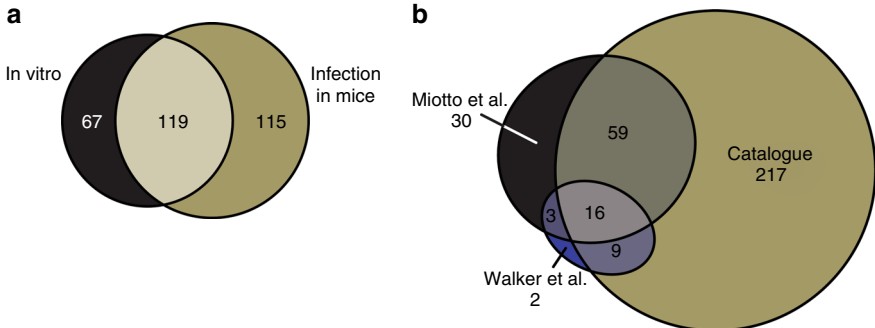

**Fig. 3** Catalog of resistant amino acid substitutions after pyrazinamide treatment in vitro and during infection in mice. **a** Comparison of enriched non-synonymous amino acid substitutions identified after pyrazinamide treatment in vitro and during infection in mice. **b** Comparison of pyrazinamide resistant non-synonymous amino acid substitutions from this study (Catalog) and from clinical isolates (Miotto et al.[32] and Walker et al.[47])

unique non-synonymous amino acid substitutions (Supplementary Data 2). An additional 210 non-synonymous amino acid substitutions whose abundance decreased after PZA selection were also observed (Supplementary Data 1 and 2).

**Selection of pyrazinamide resistant substitutions in mice.** It is unknown whether in vitro PZA selection faithfully recapitulates susceptibilities observed during infection[46]. We therefore subjected our *pncA* mutant library to PZA selection in mice. We infected BALB/c mice with the pooled *pncA* mutant library by tail-vein injection, resulting in an average implantation of $5.97 \times 10^5$ colony-forming units (CFU) in the spleen (Fig. 2c), corresponding to >300-fold library coverage. The average CFUs in the spleens increased over 42 days in the mock control group (phosphate-buffered saline (PBS) treatment) but declined from day 21 in the PZA treated mice, although the difference was not statistically significant (Fig. 2c).

We identified a total of 320 enriched SNPs that were positively selected by PZA after either 21 or 42 days of treatment corresponding to 251 unique non-synonymous amino acid substitutions (Supplementary Data 1 and 2). For those enriched after both 21 and 42 days of PZA treatment the degree of selection was on average higher after 42 days of treatment, indicating ongoing selection (Supplementary Fig. 8). An additional 219 non-synonymous amino acid substitutions whose abundance decreased after PZA treatment were also observed (Supplementary Data 1 and 2). Importantly, after 42 days of mock treatment, no amino acid substitutions were significantly increased in abundance relative to the implantation library demonstrating *pncA* SNPs have minimal fitness costs during infection and positive selection only occurred in the presence of drug (Supplementary Fig. 9).

**Enriched mutations confer pyrazinamide resistance.** To determine if enrichment during selection in our screens correlated with PZA susceptibilities we isolated single clones from our *pncA* mutant library and assayed their MICs to PZA using the BD BACTEC MGIT 960 system (Supplementary Table 1). We found that 100% of clones that were enriched either in vitro or during infection in mice had an elevated MIC (>4-fold increase compared to the Comp control). Similarly, for clones not enriched in either screen, 90% were found to be susceptible. These results demonstrate that our PZA susceptibilities predicted from both screens accurately identifies substitutions that confer resistance to PZA.

**Catalog of pyrazinamide resistant amino acid substitutions.** Having confirmed that enrichment in our screens accurately predicts PZA resistance we constructed a comprehensive catalog

of amino acid substitutions that result in PZA resistance (Supplementary Data 2). We compared amino acid substitutions enriched in vitro to those enriched after infection in mice (Fig. 3a). After exclusion of 26 amino acid substitutions that were enriched in one screen but depleted in the other, we found a high concordance. However, more resistant substitutions were found only after infection in mice than after selection in vitro, suggesting PZA selection in vitro does not fully recapitulate sensitivities observed during infection, with greater selective pressure observed during infection in mice.

In total, of a possible 977 non-synonymous amino acid substitutions resulting from a SNP, 301 (31%) were classified as conferring PZA resistance based on enrichment after in vitro and/or mouse selection. A further 310 (32%) were depleted in one or both selections and we can be confident these do not confer resistance (Supplementary Data 2). The remaining 366 non-synonymous amino acid substitutions, not selected after PZA treatment in our screens, are also likely to be susceptible based on our validation of individual clones (Supplementary Table 1). However, it is possible that some of these substitutions will confer a weak PZA resistance phenotype not apparent using conventional in vitro DST. Importantly, of 37 possible nonsense mutations 32 were classified enriched while 0 were depleted. As nonsense mutations are expected to confer PZA resistance, we can be confident in our selection criteria. Therefore, while many PncA amino acid substitutions confer PZA resistance, the majority of mutations do not.

We then compared our catalog to mutations that have previously been associated with PZA resistance in clinical isolates. Such an analysis is problematic because of variability in the genetic background of clinical isolates and the inherent limitations in MGIT DST across laboratories. Nevertheless, we compared our results to two recent retrospective studies[32, 47] where PZA susceptibilities were carried out by global reference laboratories. Despite differences between these two studies, a total of 84 (71%) PZA resistant non-synonymous amino acid substitutions found in clinical isolates were similarly cataloged PZA resistant in this study (Fig. 3b). However, our study identified an additional 217 resistant conferring substitutions, highlighting the comprehensiveness of our catalog.

**Mechanisms of pyrazinamide resistance.** Although we, like others[11, 31–33], found PZA resistance-conferring substitutions occur throughout the entire length of PncA (Fig. 4a), many other substitutions do not produce resistance to PZA. What then, is the mechanism of resistance? Mutations that disrupt the active site of PncA are expected to result in PZA resistance and we do find many substitutions in the catalytic triad[48] (amino acids 8, 96, and 138) and iron coordinating residues[48] (amino acids 49, 51, 57,

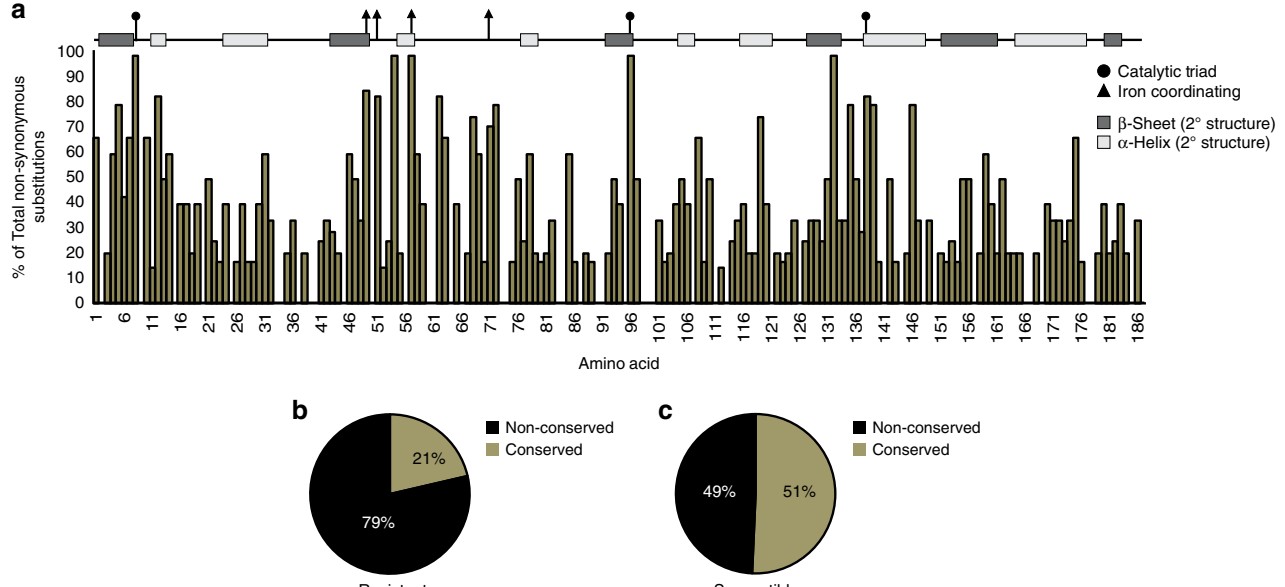

**Fig. 4** Pyrazinamide resistant amino acid substitutions occur throughout PncA and are enriched for non-conserved amino acid substitutions. **a** The proportion (%) of non-synonymous amino acid substitutions represented in the *pncA* library that were pyrazinamide resistant at each PncA amino acid. Amino acids corresponding to the catalytic triad are marked with a *circle*. Amino acids responsible for iron coordination are marked with a *triangle*. **b** Percentage (%) of non-conserved (*black*) and conserved (*brown*) pyrazinamide resistant amino acid substitutions. **c** Percentage (%) of non-conserved (*black*) and conserved (*brown*) pyrazinamide susceptible amino acid substitutions

and 71). However, resistance-conferring substitutions are also found in domains not predicted to play a role in catalytic activity. We hypothesized that alterations in protein structure may lead to decreased protein abundance and reduced activation of PZA. In support of this we found a greater percentage of cataloged resistant substitutions are non-conservative substitutions compared to susceptible substitutions (Fig. 4b, c). To experimentally test this hypothesis, we measured protein abundance and enzymatic activity in isolated single clones from our *pncA* mutant library.

We used mass spectroscopy to quantify the abundance of PncA relative to two housekeeping proteins, RpoB (Fig. 5a) and DnaK (Supplementary Fig. 10). We found that 19 of 23 resistant mutants had decreased PncA protein levels compared to WT. In general, susceptible strains had levels of PncA similar to those of WT cells. These results indicate that the majority of resistance-conferring substitutions tested result in decreased protein abundance.

Three amino acid substitutions (C138G, C138S, and D8N) had comparable levels of PncA to WT, but are cataloged as resistant. These substitutions all occur within the catalytic triad, suggesting they could impact enzyme activity. To confirm this we used a modified quantitative Wayne assay[49] to determine catalytic activity. Of the three PZA resistant clones tested all had reduced PZase activity, relative to the Comp strain (Fig. 5b). In contrast, susceptible isolates had equivalent enzymatic activity. This indicates, for the subset of PncA mutations tested, that loss of catalytic activity as well as reduced protein levels can mechanistically contribute to PZA resistance.

The reduction in protein abundance caused by non-active site substitutions suggests interference with protein folding and PncA stability. Amino acids buried within the interior of the protein, which have low solvent accessibility, will be the most critical for proper protein folding[50]. To test this, we investigated the predicted solvent accessibility[51], obtained using the previously reported *Mtb* PncA crystal structure[48], of amino acids that confer resistance. We found a consistent negative correlation ($r = -0.53$)

between the solvent accessibility and the frequency of resistance-conferring substitution at each amino acid (Fig. 5c and Supplementary Fig. 11). Furthermore, all PncA amino acids at which no substitution resulted in PZA resistance generally map to surface exposed residues that do not overlap with the active site (Fig. 6a, b). Substitutions at these amino acids are therefore predicted to have little or no effect on protein folding or catalytic activity. Combined, these results suggest that resistance results largely from a loss of enzymatic activity and/or protein abundance, which can be empirically determined but not easily predicted from the primary sequence.

## Discussion

PZA resistance prevents the use of this critical drug in shortening TB treatment, both in current therapy and possible future regimens. Unfortunately, the current methods of testing for PZA resistance are phenotypic, which are difficult and unreliable. In fact, it has been proposed that any non-synonymous mutation within *pncA* can be considered a PZA resistant allele[52]. In this study, we find that this is not the case. We used saturating mutagenesis of *pncA* to systematically screen for amino acid polymorphisms that confer resistance to PZA. While many mutations are associated with resistance to PZA, most substitutions were susceptible. Nonetheless, our results have significantly expanded the repertoire of PZA resistance-conferring mutations, as most of these mutations have not been observed in clinical isolates.

Importantly, our catalog of PZA susceptibilities shows that a molecular sequencing based diagnostic of *pncA* could be used to both rule in and rule out patients as candidates for successful PZA therapy, a key development in determining and designing successful treatment regimens. This study generates a comprehensive reference for interpreting the PZA susceptibility of *pncA* alleles and is an essential advance in developing rapid molecular diagnostics for PZA resistance.

Our assays relied on statistical tests to evaluate PZA susceptibility, so we, like others, cannot rule out all false negative and, to

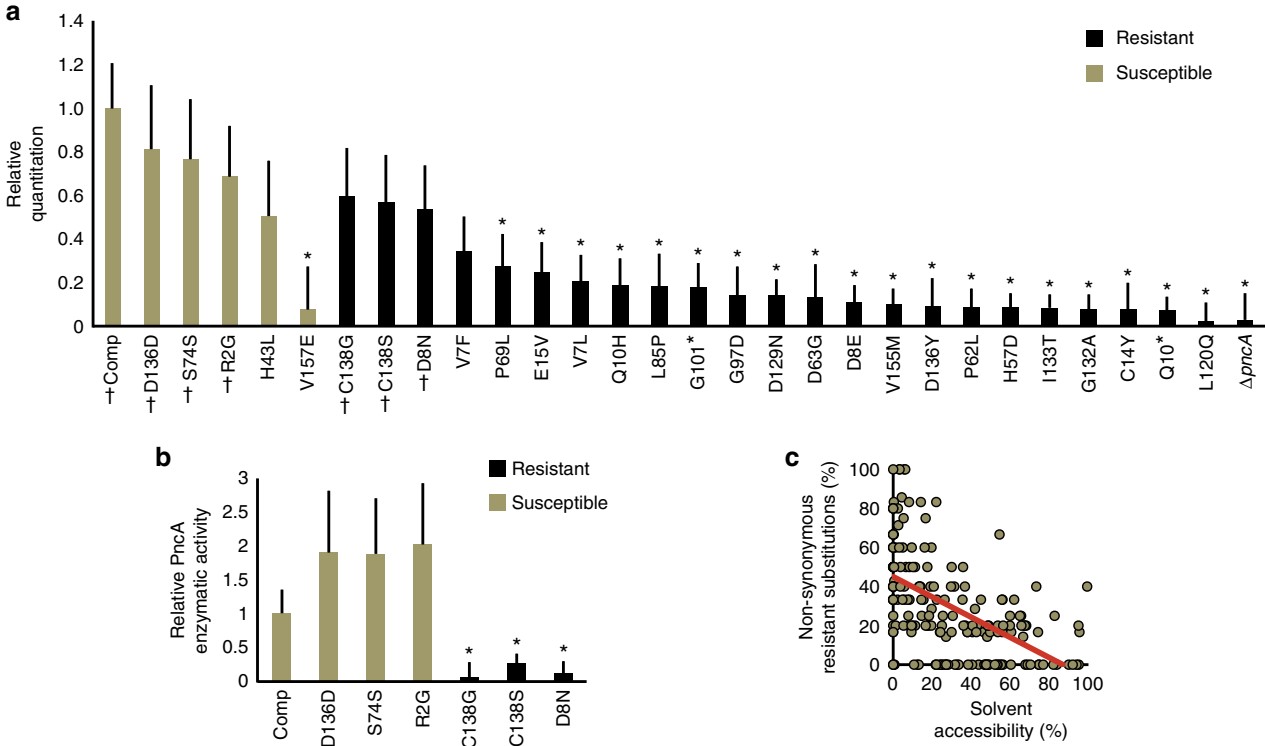

**Fig. 5** PncA protein abundance and catalytic activity result in pyrazinamide resistance. **a** Quantitation of PncA protein abundance relative to RpoB in cataloged pyrazinamide susceptible (*brown*) and resistant (*black*) isolates determined using mass spectrometry. Protein abundance in each mutant strain is relative to the Comp strain. *$P \leq 0.01$ (*t*-test, two-sided). † corresponds to isolates tested for enzymatic activity. The mean fold-change of three biological replicates for each mutant relative to the mean of 10 biological replicates for Comp is shown. *Errors bars* represent the standard deviations derived from the propagation of error using the quotient of the coefficient of variation. **b** Mean PncA catalytic activity relative to the Comp strain is shown. *$P \leq 0.05$ (*t*-test, two-sided). *Error bars* represent the standard deviations derived from three biological replicates for each strain by the propagation of error using the quotient of the coefficient of variation. **c** Correlation between the solvent accessibility (%) and the proportion (%) of all possible non-synonymous amino acid substitutions that confer pyrazinamide resistance at each amino acid in PncA. Linear trend line is shown in *red* ($r = -0.53$)

a lesser extent, false positive calls. However, assays of individual mutants from our study strongly suggest that our predictions for resistance-conferring alleles are reliable. More importantly, our in vitro derived library enables us to test all possible mutations for PZA susceptibilities in an isogeneic strain background, and exclude the contribution of other resistance genes. This eliminates variability in genetic background and growth rates that significantly complicate studies using clinical isolates. If other genes do play a role, sequencing of *pncA* may not predict all PZA resistance observed in clinical isolates. For example, both *rpsA*[53] and *panD*[54, 55] may be important in the mechanism of PZA action. However to date, few resistance mutations in either *rpsA* or *panD*[54, 56–58] have been reported in clinical isolates, and, at this point, they likely account for a minority of PZA resistance.

The importance of understanding the mechanisms of antibiotic resistance cannot be understated. Strikingly, mutations conferring resistance to PZA occur throughout the entire length of PncA[11, 31–33]. The majority of the substitutions we evaluated using proteomics were associated with reduced PncA protein levels which may account for the widespread intragenic distribution of drug resistance-conferring mutations. Those with little or no effect on protein abundance did map to active site regions and had reduced enzyme activity indicating disruption of catalysis is also a mechanism of resistance. Crystallographic studies will be needed to confirm these findings. Furthermore, despite multiple selections some codons, mapping almost exclusively to surfaced exposed residues, failed to enrich for any PZA resistant substitution. Interestingly, while it is known that *pncA* is non-essential[37–41], its in vivo function is poorly understood.

Consistently, we observed no generalizable fitness defects for *pncA* mutations in the absence of PZA treatment. However, given the synergistic drug interactions that PZA exhibits[4, 59, 60], it would be intriguing to explore the effects of PncA resistant substitutions generated in this study, on these interactions. These results underscore the value of a comprehensive mutational and functional analysis of drug resistance mechanisms.

We assessed the direct correlation between PZA susceptibility in vitro and during infection in mice. Reassuringly, amino acid substitutions found to be resistant in vitro were also identified in our mouse model. However, many more PZA resistance substitutions were detected after treatment in mice than observed in vitro. When individual mutants, classified as resistant in the mouse screen, were assayed using standard DST we were able to confirm they were truly resistant. Studies have shown that the activity of PZA is augmented by host immunity during treatment in a mouse[61] and is bactericidal in a three dimensional cellular model in contrast to the bacteriostasis seen in vitro[62]. A stronger antibiotic selective pressure due to the enhanced activity of PZA in the host is therefore the most likely reason why we identified more mutations in the mouse. It is also conceivable, but unlikely, that phenotypic resistance or antibiotic tolerance may be implicated. It is clear from our single-colony validations that resistance to PZA is directly linked to specific genetic mutations contained within different alleles of *pncA*.

Our murine model does not perfectly replicate infection in a human, and we had to recover bacilli from the spleen to ensure representation of our library after inoculation. Nevertheless, PZA treatment in mice is certainly closer to human chemotherapy

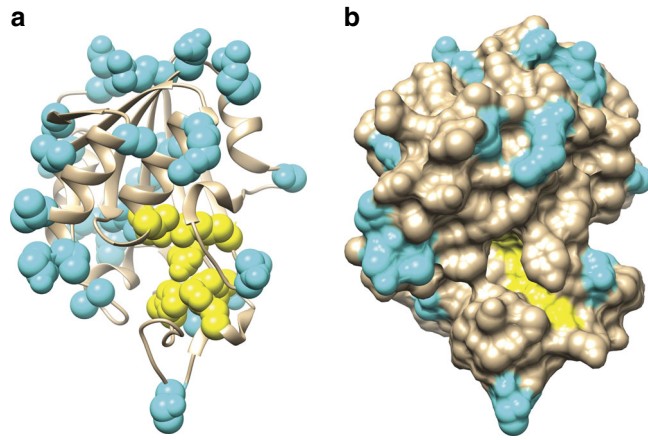

**Fig. 6** Amino acid substitutions to which no substitution resulted in pyrazinamide resistance ($n = 29$) map to surface exposed regions of PncA. **a** Ribbon diagram of the PncA crystal structure[48]. **b** Surface representation of the PncA crystal structure[48]. *Cyan* represent amino acids to which no substitution resulted in pyrazinamide resistance. *Yellow* are active site and iron coordinating amino acids

than in vitro treatment in acid media. These results thus highlight the limitations of only using an in vitro assay for predicting PZA resistance in clinical samples. For these reasons we are confident that our comprehensive catalog of PncA substitutions, generated from both in vitro and mouse infection screens, does predict clinically relevant PZA resistance. For most anti-TB drugs, and certainly for PZA, there are few data associating in vitro antibiotic susceptibility of specific mutations with treatment outcomes in humans[63]. Further studies in animal models and human cohorts are therefore needed to determine how individual mutations influence treatment efficacy particularly in the context of combination therapy. Our comprehensive mutational and screening approach should stand as a paradigm for identifying resistance mutations. The dual approach of DST both in vitro and in mice may represent a new model to comprehensively screen for the most clinically relevant mutations that includes a readout of therapeutic success.

## Methods

**Bacterial strains, plasmids, and media**. We used *E. coli* DH5α and *Mtb* H37Rv (sAY101) in this study. Strains, plasmids, and primers are listed in Supplementary Figs. 2–5) in this study. Unless otherwise noted, *Mtb* strains were grown using Middlebrook 7H9 (0.2% glycerol and 0.05% Tween 80) or 7H10 agar (0.5% glycerol) (Difco, BD) and supplemented with OADC (BBL, BD). *Mtb* from the in vivo screen were plated on Middlebrook 7H11 (0.5% glycerol) (Difco, BD) and supplemented with OADC (BBL, BD), 200,000 IU l$^{-1}$ polymixin B, 10 mg l$^{-1}$ amphotericin B, 50 mg l$^{-1}$ cabenicillin, and 20 mg l$^{-1}$ trimethoprim. Zeocin was used at 50 µg ml$^{-1}$ and 20 µg ml$^{-1}$ for *E. coli* and *Mtb*, respectively. Hygromcyin was used at 50 µg ml$^{-1}$ for *Mtb*. All strains were grown at 37 °C.

WT *pncA* and MOP[45] were PCR amplified using ANY_P55/ANY_P57 and ANY_P149/ANY_P150, respectively, and sequentially cloned into ZeoR, L5 integrating vector pAY59 using *ClaI/PciI* and *ClaI/NotI*, respectively, to generate pAY107 then pAY108 (Comp). The vector control (pAY111) was generated by PCR amplification of Emerald using ANY_P147/ANY_P148 and cloning into pAY107 using *NotI/XbaI*. GeneMorph II Random Mutagenesis Kit (Agilent Technologies) was used according the manufactures protocols using ANY_P153/ANY_P154. Amplicons were subsequently cloned into pAY112 using *NotI/XbaI* to generate our library of *pncA* SNPs. 50,000 *E. coli* clones were harvested by scraping to create our pooled *pncA* SNP library (pAY230). Approximately 34,000 *Mtb* transformants were scrapped to constitute sAY260.

**RT-qPCR**. RNA was extracted from 25 ml, mid-log phase (OD$_{600}$ ≈ 0.5–0.6) cultures. Pelleted cells were suspended in 1 ml of TRIzol (Ambion, Life Technologies). Five repeated cycles of 1 min bead-beating using MagnaLyser (Roche) and 1 min at −20 °C with 100 µl, 0.1 mm sterile zirconia/silica beads (Biospec) was performed followed by chloroform extraction and ethanol precipitation. RNA was DNase (Fermentas) treated prior to cDNA synthesis using iScript Advanced cDNA

Synthesis Kit (Bio-Rad) according to the manufactures protocol. qPCR was performed on CFX 96 Real Time PCR detection system (Bio-Rad) using iTaq Universal SYBR Green Supermix (Bio-Rad) and KM_P35/KM_P36 and sigA_F/sigA_R for *pncA* and *sigA*, respectively. All samples were performed in biological triplicate, internally normalized to *sigA*, and the mean fold-change and standard deviations evaluated relative to WT (sAY101).

**In vitro drug-susceptibility screening**. PZA and pyrazinoic acid susceptibility were evaluated using the BD BACTEC MGIT 960[12] system with manufacturer supplied PZA medium/supplement and BBL MGIT medium/growth supplement, respectively. Mycobacterial growth indicator tubes (MGITs) were inoculated with 50,000 CFU directly from a thawed, titered glycerol stocks. PZA (Sigma Aldrich) or pyrazinoic acid (Sigma Aldrich) was added to a final concentration of 0, 1, 4, 20, 100, 500, or 1000 µg ml$^{-1}$. The mean TTP and standard deviations of three biological replicates relative to the no-drug control is reported.

The sequential PZA susceptibility screen was performed using the BD BACTEC MGIT 960[12] system with manufacturer supplied PZA medium/supplement containing 0, 4, 20, 100 or, 500 µg ml$^{-1}$ PZA (Sigma Aldrich). The first selection was inoculated with 50,000 CFUs directly from thawed, titered glycerol stocks of sAY260. After ≤24 h of individual MGITs reaching their TTP, bacterial cells were collected by centrifugation and suspended in 1 ml 7H9 broth. An aliquot of 30 µl was re-inoculated to a new MGIT containing the same PZA concentration. The remaining bacteria were pelleted by centrifugation and frozen at −20 °C. This was repeated for a total of 6 selection rounds. Genomic DNA from frozen bacterial pellets of each selection round was extracted using the Hain GenoLyse Kit according to the manufactures protocol. Three biological replicates were performed per drug concentration.

**Drug-susceptibility testing**. PZA and pyrazinoic acid DST was performed using the BACTEC MGIT 960[12] system with manufactured supplied PZA medium/supplement and the five place antimicrobial susceptibility test (AST) set carrier and protocol with the following modifications. PZA concentrations ranged from 0.2 to 500 µg ml$^{-1}$. Pyrazinoic acid concentrations ranged from 1 to 1000 µg ml$^{-1}$. PZA and pyrazinoic acid containing MGITs were inoculated with 25,000 CFU directly from thawed, titered glycerol stocks. No-drug MGITs were inoculated with 2500 CFUs directly from a thawed, titered glycerol stocks. After the AST no-drug control MGIT reached positivity, the MICs were recorded as the lowest drug concentration with a growth index <100.

**Pyrazinamide screening in mice**. sAY260 was grown to OD$_{600}$ = 1.0 then washed and suspended in PBS. A total of 6–8 week-old female BALB/c mice (~20 g each; supplied by the University of KwaZulu-Natal) were infected by tail-vein injection with 10$^6$ CFUs. Mice were arbitrarily randomized into two cohorts, receiving 150 mg kg$^{-1}$ PZA or PBS by oral gavage 5 days per week for 42 days, four days after infection (day 0). Five mice from each cohort were randomly selected for sacrifice and the spleens harvested on day −3, day −0, day 21 and day 42 to determine the CFUs and quantify *pncA* SNPs. Five mice per time-point provided sufficient statistical comparisons within each group and to mitigate loss of samples due to contamination. No mice or samples were excluded from the analysis. The authors were not blinded to cohort allocation or assessment of the outcomes. Spleens were homogenized in 2 ml PBS using a mortar and pestle. CFUs are reported as mean and standard deviations of total bacterial burden in the organ. The coefficient of variation (CV) between control groups at day −3 and day 0 were highly similar, indicating technical reproducibility in infection. The CV between PZA and mock treated mice increases after 21 and 42 days, consistent with variability in how mice innately control the initial burden of infection. Greater than 10,000 estimated CFU per mouse were scrapped, pooled, and frozen for guanidine thiocynate (GTC) genomic DNA extraction as described[64] with the following modifications. Two chloroform extractions and ethanol washes were performed. All mouse procedures were performed at African Health Research Institute (AHRI; formed by the union of KwaZulu-Natal Research Institute for TB and HIV [K-RITH] and the African Centre on 1st October 2016) and were reviewed, with ethical approval (Ref 116/14/ Animals "In vivo identification of PZA resistant SNPs of Mtb"), by the Animal Research Ethics Committee of the University of KwaZulu-Natal.

**Sequencing**. Single-colony *pncA* clones from either *E. coli* or *Mtb* libraries were PCR amplified using ANY_P58/ANY_P59 and sequenced using the ABI Big Dye Terminator Kit, with either ANY_P58 or ANY_P59, using ABI Prism 3130 sequencer according to manufacturers. After selection in vitro and infection in mice, *pncA* was PCR amplified using ANY_P158/ANY_P159 from bulk genomic DNA. Libraries for next generation sequencing (NGS) were prepared using standard Illlumina genomic DNA sequencing sample preparation protocol (Illumina, Inc.), with barcodes for multiplexing. Shearing was performed using a Covaris E220 (Covaris, Inc.). Samples were sequenced in paired-end mode (125 + 125 bp) on an Illumina 2500 sequencer. On average 2.9 million pairs of reads were collected for each sample and mapped to the *Mtb* H37Rv *pncA* gene using the Burroughs-Wheeler Aligner program[65]. Pairs of reads with two or more mismatches or containing indels were discarded. The number of SNPs at each position was tabulated and individual SNP frequencies were calculated by dividing the count by the total number of reads covering that site in each data set. The frequency of SNPs

from a sequenced unselected WT *pncA* clone was subtracted from the observed frequencies in each sample. Enrichment was calculated as the log2 mean fold-change for each PZA selection relative to the corresponding no-drug control of the same round (in vitro) or treatment length (mouse infection).

Statistical significance was determined by fitting a negative binomial distribution to each replicate of normalized read counts in each condition, using empirical Bayes estimation of variances, and comparing the differences in scaled means using a Wald test, as implemented in DESeq2[66]. The resulting *P*-values were adjusted for multiple comparisons using the Benjamini–Hochberg procedure. For each in vitro PZA concentration, SNPs exhibiting ≥2 fold-change (adjusted $P \leq 0.05$) in two or more selection rounds were identified. Mutations were then collated across all PZA concentrations and compressed to their respective amino acid substitutions. Mutations identified as both enriched and depleted at any point were ignored. In mice, SNPs exhibiting ≥2 fold-change (adjusted $P \leq 0.05$) after either 21 or 42 days of PZA treatment were identified. Mutations were then collated across both selection days and compressed to their respective amino acid substitutions. Mutations identified as both enriched and depleted at any point were ignored. Any amino acid substitution enriched or depleted after either in vitro and/or mice selection, were cataloged as PZA resistant or susceptible, respectively.

**PncA proteomics.** The proteome of sAY245 (Comp) was uniformly $^{13}C$ labeled by growing cells to $OD_{600} = 1$ in a modified Hartmans-de Bont (HDB) media: 3.4 μM EDTA, 49 μM MgCl₂•6H₂O, 0.7 μM CaCl₂•2H₂O, 0.08 μM NaMoO₄•2H₂O, 0.17 μM CoCl₂•6H₂O, 0.62 μM MnCl₂•2H₂O, 0.7 μM ZnSO₄•7H₂O, 1.8 μM FeSO₄•7H₂O, 0.08 μM CuSO₄•5H₂O, 890 μM K₂HPO₄, 708 μM NaH₂PO₄, 15 mM (NH₄)₂SO₄, 27.4 mM glycerol-$^{13}C_3$, and 0.05% tyloxapol.

Lysates for $^{13}C$ labeled sAY245 and unlabeled samples were prepared from 30 ml cultures grown to mid-log phase ($OD_{600} = 0.7$). Pelleted cells were washed three times with 10% glycerol and suspended in PBS containing protease inhibitor cocktail (Roche) and 100 μl, 0.1 mm sterile zirconia/silica beads (Biospec). 4 repeated cycles of 1 min bead-beating at 7000 rpm using MagnaLyser (Roche) and 5 min at −20 °C was used to rupture cells. A final concentration of 2% SDS was added and samples were incubated at 65 °C for 30 minutes. Lysates were cleared twice by centrifugation, passed through 0.2 μm Corning Costar SpinX columns (Sigma Aldrich), and stored at −80 °C.

Lysates from $^{13}C$ labeled sAY245 and unlabeled samples were mixed 1:1 in total volume of 200 μl. Proteins were extracted by the addition of 600 μl methanol, 150 μl chloroform, and 450 μl H₂O and centrifugation. Proteins were then precipitated by addition of 400 μl methanol to the organic phase of each sample followed by centrifugation. Protein was suspended in 100 μl 6 M urea, 25 μl 200 mM DTT, and 20 μl 200 mM iodoacetamide. Samples were digested overnight with 4 ng trypsin (Promega) in total volume of 1 ml. A total of 10 μl acetic acid was added and each sample was column purified using Sep-Pak Vac 6cc 500 mg C18 cartridges (Waters, WAT043395), washed with 10 ml 2% acetonitrile with 0.1% formic acid and eluted with 2 ml 65% acetonitrile with 0.1% formic acid. Columns were cleaned and equilibrated, prior to the addition of each sample, by sequential addition of 5 ml acetonitrile, 5 ml 65% acetonitrile with 0.1% formic acid, and 10 ml 2% acetonitrile with 0.1% formic acid. Eluted peptides were dried by vacuum centrifugation, suspended in 50 μl 2% acetonitrile with 0.1% formic acid and stored at 4 °C.

Unique *pncA*, *rpoB*, and *dnaK* peptides (Supplementary Table 7) were targeted using a Q Exactive MS coupled to a Dionex RSLC 3000 nano LC system (Thermo Scientific), using a 30 min gradient and mobile phase from 1% acetonitrile and 0.1% formic acid to 50% acetonitrile and 0.1% formic acid, with a regeneration phase to clean up and equilibrate the column. Separation was carried out on a 15 cm by 75 μM ID fused silica column, backed with 3 μM C18 stationary phase, and a flow rate of 0.300 μl min⁻¹. The nano-spray source was operated at a capillary temperature of 275 °C, an emitter voltage of 1.6 Kv and S-lens voltage of −55 V. A targeted-MS2 method was employed with an AGC target of 1.0e⁵ ions, an isolation width of 2 m z⁻¹ and an HCD setting of 27.

Spectral traces were quantified using Skyline proteomics software. The area-under-the-curve (AUC) for each peptide transition was exported using the MS1 probe plug-in. Any transitions containing the mutation of interest or with an AUC = 0 were discarded. For each sample, the quantified light (L) AUC was normalized to the $^{13}C$-heavy (H) AUC for each transition (L/H). The mean L/H for all *pncA* peptides was then normalized to the mean L/H for all *rpoB* or *dnaK* peptides across all biological replicates. Three biological replicates each per mutant and 10 biological replicates for sAY254 (Comp) was performed. The fold-change and standard deviations of each mutant relative to sAY254 (Comp) is reported.

**Wayne assay.** A modified quantitative Wayne assay[49] was used to determine the enzymatic activity of PncA. Cultures were grown, in triplicate, to $OD_{600} = 0.6–0.8$ and treated with 500 μg ml⁻¹ PZA for 48 h. A 500 μl sample with 100 μl 0.1 mm sterile zirconia/silica beads (Biospec) underwent four repeated cycles of 1 min bead-beating using MagnaLyser (Roche) and 1 min at −20 °C. An aliquot of 10 μl 20% ferrous ammonium sulphate (Sigma) was added to 100 μl cleared lysate. The reaction was terminated by addition of 890 μl 0.1 M glycine-HCl (pH 3.4). The absorbance at 460 nm was quantified using a BioRad SmartSpec Plus. The absorbance of each 48 h PZA treated sample was normalized to the absorbance prior to PZA treatment and scaled by both bulk protein (BCA Protein Assay Kit, ThermoFisher Scientific) and PncA (proteomics) levels. The relative PncA enzymatic activity is reported as the mean and standard deviation relative to sAY245 (Comp).

**Clinical isolate comparison.** PZA DST in clinical isolates was obtained from Miotto et al.[32] and Walker et al.[47] PZA resistant non-synonymous SNPs were extracted based on sequencing of *pncA* and associated drug-susceptibility to PZA as determined in each study. Indels, multi-nucleotide polymorphisms, and discrepant (both resistant and susceptible) PZA drug-susceptibility calls were removed.

**PncA protein structure modeling.** PncA amino acids (29) to which no substitution resulted in PZA resistance were mapped to the previously reported *Mtb* PncA crystal structure[48].

**Statistical analysis.** Sample sizes were chosen as the minimum number of replicates (based on limited resources and time) for statistical analysis and are indicated for each experiment. Unless otherwise indicated, no samples were preferentially excluded from any analysis and all data are presumed to meet assumptions of each statistical test with variance between groups similar.

**Data availability.** The raw Illumina sequencing reads generated in this study can be ascertained at https://www.ncbi.nlm.nih.gov/ using the SRA accession number SRP093213. The authors declare that all other data supporting the findings of this study are available within the article and its Supplementary Information files, or are available from the authors upon request.

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

## Acknowledgements

We would like to kindly thank: Helena Boshoff and Valerie Mizrahi for providing the *Mtb pncA*-null strain, Chivonne Moodley for management of the animal facility, and all lab members for their ideas, constructive criticism, and helpful comments. This work was funded by U19AI109755 to E.J.R. and R21AI116142 to E.J.R. and A.S.P.

## Author contributions

A.N.Y. and K.M. contributed intellectual insight and performed all experiments. Y.L. assisted in assessment of resistant mutations in clinical isolates. J.C.S. assisted with

Illumina sequencing. T.R.I. contributed primary analysis of sequencing data and intellectual insight. J.H.A. developed and performed mass spectroscopy experiments. A.S.P. and E.J.R. contributed intellectual insight and guidance. A.N.Y., K.M., T.R.I., A.S.P., and E.J.R. wrote the manuscript.

## Additional information

**Competing interests:** The authors declare no competing financial interests.

