## [Peer Review File · Nature Communications]

Reviewers' Comments:

Reviewer #1:

Remarks to the Author:

This manuscript describes the use of a comprehensive mutational and screening strategy to determine the range of scope of *pncA* mutations in that confer functional PZA resistance to *Mtb*. The significance of this study is that PZA is an important component of the standard combination drug therapy that is universally used to treat drug susceptible and drug resistant TB treatment. Currently, testing for PZA resistance is not routinely done and the current tests used to screen for resistance are limited to small genomic regions. The significance of this study is that it surveyed the entire PZA resistance gene and identified all potential mutations that have the potential to confer PZA resistance to *Mtb*. A major strength of the study is that potential mutants were screened *in vitro* and *in vivo* and in addition to known mutations, this study also identified mutations that have not been previously described.

The terminology used by the authors describing PZA resistance is confusing. They refer to phenotypic PZA resistance rather than genotypic resistance. It is the understanding of the reviewer that mutations in the resistance gene as modeled here is genotypic resistance where as the expression of drug resistance as a consequence of both host and pathogen factors that contribute to the expression of drug resistance in the absence of genetic mutations (biofilm formation, poor drug penetration etc.) should technically be referred to as phenotypic resistance.

The strategy used to generate PZA resistant mutants is innovative and the study is well controlled. Among the weaknesses of the study is the lack of concordance between the *in vitro* and *in vivo* selection where there were more resistant substitutions in isolates from infected mice compared to the *in vitro* screen which may represent down-stream effects that confer phenotypic rather than classical genotypic resistance identified through the *in vitro* screen. The authors should further clarify how they interpret these data and more importantly, how to explain the results and the clinical implications. While the authors concluded that the various AA substitutions had no significant impact on fitness, the discrepancy between the *in vitro* and *in vivo* screening could be related to differences in fitness. It is still unclear to the reviewer (and other potential readers) what impact specific AA substitutions had on *in vivo* resistance given the pooled infection and spleen recovery beyond the statistical approach. This should be made clearer.

It is recommended that the authors include DnaK as well as RpoB protein levels in the quantitation of PncA via mass spectroscopy. They indicate that DnaK data was not shown but did not justify the reason, given that they concluded that a decrease in protein abundance explained the majority of the resistance-conferring substitutions.

The authors are at risk of over interpreting the data related to the 3 AA substitutions within the catalytic region of PncA to explain the mechanism of resistance, given the lack of structural conformation. Similarly, the impact AA substitutions had on protein folding and stability.

Reviewer #2:

Remarks to the Author:

Pyrazinamide is the least understood front-line drug and resistance to this drug is not completely understood. In this work the authors perform the first complete analysis of the PncA mutations that are associated with pyrazinamide resistance *in vitro* as well as *in vivo*. This is performed by performing random PCR mutagenesis on the *pncA* gene, transfer of this to a deletion mutant of *pncA* in *M. tuberculosis* followed by multiple rounds of selection *in vitro* as well as selection after high dose infection of mice. Mutations are identified across the entire gene although a large majority of those that did not occur in the residues important for catalysis were found to map to solvent accessible areas. The authors suggest that this is due to decreased protein stability

resulting in lowered expression levels and indeed find this to be the case for almost all of those tested with the exception of a few that were found to confer drastically reduced enzymatic activity. The in vivo selection is found to be more stringent than the in vitro selection which lends strong support to the utility of the PncA mutation database in providing a valuable reference guide for predicting pyrazinamidase-related resistance in human patients. The novel aspects include: 1. This is the first time that random PCR mutagenesis has been applied to determine the contributions of amino acid mutations across an entire gene known to play a role in drug activation in M. tuberculosis and would serve as a valuable model for others trying to understand the role of mutations that play a role in the mechanism of action of drugs that are difficult to study in vitro or when the mutations identified in preliminary studies do not consistently map to a resistance determining region; 2. a better understanding of why mutations map to amino acids outside of the catalytically important residues; 3. the mapping of mutations found here to those previously identified in the literature.

The work is well done and the only small comment is that the authors need to include an additional column to Table S1 mapping up the mutations identified in the published literature, which would be especially important if there is a discrepancy (thus if mutations in the literature are predicted to confer resistance and found sensitive here, it would be critical to include this).

HB

Response to Reviewers

Reviewer #1 (Remarks to the Author):

1. The terminology used by the authors describing PZA resistance is confusing. They refer to phenotypic PZA resistance rather than genotypic resistance. It is the understanding of the reviewer that mutations in the resistance gene as modeled here is genotypic resistance where as the expression of drug resistance as a consequence of both host and pathogen factors that contribute to the expression of drug resistance in the absence of genetic mutations (biofilm formation, poor drug penetration etc.) should technically be referred to as phenotypic resistance.

We agree that the terminology to distinguish genetic resistance from phenotypic resistance can be confusing. To rectify this ambiguity, we have removed our use of the phrase “phenotypic resistance” throughout the text and resorted solely to stating “resistance.” We thank the reviewer for this comment as the correction clarifies our results modeling genotypic resistance for the reader.

2. The strategy used to generate PZA resistant mutants is innovative and the study is well controlled. Among the weaknesses of the study is the lack of concordance between the *in vitro* and *in vivo* selection where there were more resistant substitutions in isolates from infected mice compared to the *in vitro* screen which may represent down-stream effects that confer phenotypic rather than classical genotypic resistance identified through the *in vitro* screen. The authors should further clarify how they interpret these data and more importantly, how to explain the results and the clinical implications.

We thank the reviewer for this suggestion to improve the interpretation of our results, and we have expanded our discussion to present them more clearly.

The reviewer highlights one of the important findings of our study, that we identified more resistant mutations in our mouse screen compared to our *in vitro* screen. The activity of pyrazinamide *in vitro* is poor. It was for this reason that we also conducted our screen in mice, conditions that we feel better approximates the activity and efficacy of pyrazinamide in humans. We believe that the increased number of resistant mutations identified in the mice screen was due to a greater selective pressure exerted under conditions in which pyrazinamide is more active. We have elaborated on this explanation on page 15, line 327. In short, studies have demonstrated that pyrazinamide more effectively kills *M. tuberculosis* in a 3-dimensional granuloma model than *in vitro*. Furthermore, host immune systems are believed to contribute to pyrazinamide activity. As a result, we believe that host-derived selection pressures likely contribute to the increased pyrazinamide susceptibility we observed in our mouse screen relative to the *in vitro* screen.

The reviewer however, raises an interesting alternative explanation to the one stated above. Specifically that the increased selection is not due to classical genetic resistance but rather non-genetic phenotypic resistance. We think this is unlikely because the mutations that were selectively enriched in the mouse screen proved to be resistant when the susceptibility was tested *in vitro*. It is clear from our screens and validation assays that resistance to pyrazinamide is directly linked to specific genetic mutations contained within different alleles of *pncA*. Needless-to-say, we have addressed the reviewers explanation on page 15, line 333, which we feel further strengthens the importance screening for antibiotic resistance both *in vitro* and in an animal model.

For the above reasons we are confident that the mutations selected after infection in mice are truly resistant – a key objective of our study. Ultimately the full clinical impact of individual *pncA* alleles to pyrazinamide resistance needs to be determined using multi-drug animal infection models and with human studies. This is beyond the scope of our manuscript but we have expounded on the importance of these experiments on page 15, line 342.

3. While the authors concluded that the various AA substitutions had no significant impact on fitness, the discrepancy between the *in vitro* and *in vivo* screening could be related to differences in fitness.

To directly assess the impact of the mutations on fitness we included a no drug or mock treatment control in both our *in vitro* and mouse screens, respectively. When we compared the input or implantation library to the no drug or untreated mice libraries we found no evidence of selection (Supplemental Fig 6 & 9). We believe this is strong evidence that in the absence of pyrazinamide there were no fitness costs that could explain why we identified more resistance mutations after selection in mice. However, we did not show the data relating to the untreated mice libraries in the original submission, but agree they are important. We therefore have added a Supplementary Fig. 9 (page 8, line 179) showing the distribution in the mutational fold-change between day 42 and implantation of mock treated animals.

4. It is still unclear to the reviewer (and other potential readers) what impact specific AA substitutions had on *in vivo* resistance given the pooled infection and spleen recovery beyond the statistical approach. This should be made clearer.

We thank the reviewer for highlighting this lack of clarity with regard to the impact of selected *pncA* mutations on pyrazinamide resistance in mice. We have expanded our discussion to address this (page 15, line 327; page 15; line 337; page 15, line 342)

Our approach to verifying the pyrazinamide susceptibilities of *pncA* mutations identified after selection in mice was to assay for pyrazinamide resistance of single clones using the drug susceptibility test approved for defining clinical resistance (MGIT). When we did this we found a very high concordance between mutations predicted to be resistant in the mouse and their level of resistance determined using a susceptibility test (see Supplementary Table 3). Given this we are confident they confer resistance as defined by clinical microbiological standards and have qualified this on page 9, line 189)

The rationale for pooling was to avoid creating and testing all individual mutants which is not possible due to the total number of mutants (>1000). While it is conceivable that screening of pooled mutants could produce artefactual results, we suspect this would be equally likely to have occurred *in vitro* as during infection in mice. We therefore feel this could not explain the increased number of mutations identified in the mouse infection screen.

Many TB animal models perform selection in the lungs of infected animals. Because of the complexity of our *pncA* mutant library, it was not possible to adequately implant a large enough inoculum in the lungs of infected animals to ensure complete representation of the library. It is possible that the activity of pyrazinamide in the lungs is different from that in the spleens, but a recent study has shown that even in a cellular model pyrazinamide has enhanced activity. However we agree it is a limitation of our study and we have mentioned this in the discussion (page 15, line 337)

5. It is recommended that the authors include DnaK as well as RpoB protein levels in the quantitation of PncA via mass spectroscopy. They indicate that DnaK data was not shown but did not justify the reason, given that they concluded that a decrease in protein abundance explained the majority of the resistance-conferring substitutions.

As suggested by the reviewer we have included the DnaK normalized samples as Supplemental Fig. 10 (page 11, line 224).

6. The authors are at risk of over interpreting the data related to the 3 AA substitutions within the catalytic region of PncA to explain the mechanism of resistance, given the lack of structural conformation. Similarly, the impact AA substitutions had on protein folding and stability.

We thank the reviewer again for identifying an area where the interpretation of our data could be improved. To address these concerns we have qualified our data in both the results section (page 11, line 255 and page 12, line 268) and the discussion section (page 14, line 5). Here we indicate that the observed effects reflects only the amino acid substitutions actually tested. We also acknowledge we have not defined crystallographic structures for the mutations tested and our predictions instead rely on modeling the mutations onto the structure. We have added this qualification to the discussion section (page 14, line 3`3). We have also referenced how our structure based predictions are based on an experimentally derived crystal structure of *M. tuberculosis* PncA (page 12, line 261)

Reviewer #2 (Remarks to the Author):

The work is well done and the only small comment is that the authors need to include an additional column to Table S1 mapping up the mutations identified in the published literature, which would be especially important if there is a discrepancy (thus if mutations in the literature are predicted to confer resistance and found sensitive here, it would be critical to include this).

Thank you for this important suggestion. We have added the resistant mutations published from both the Miotto et. al and Walker et. al studies referred to in this manuscript to Supplemental Table 2 which reports amino acid substitutions. In this manuscript, we compared our resistant amino acids to published resistant amino acid substitutions in the two studies. We therefore felt it was more appropriate and useful to the reader to include the amino acid resistance rather than the nucleotide resistance. Hence our inclusion of the mutations in Supplemental Table 2 rather than Supplemental Table 1 (which reports nucleotide substitutions).

Reviewers' Comments:

Reviewer #1:

Remarks to the Author:

Having reviewed the thoughtful responses to the reviewers comments and concerns as well as the revised manuscript, I am pleased with the responses. I think the manuscript reads well and is much improved in areas sections that lacked clarity.